# Genome-Wide Analysis of WRKY Transcription Factors Involved in Abiotic Stress and ABA Response in *Caragana korshinskii*

**DOI:** 10.3390/ijms24119519

**Published:** 2023-05-30

**Authors:** Jinhua Liu, Guojing Li, Ruigang Wang, Guangxia Wang, Yongqing Wan

**Affiliations:** Key Laboratory of Plants Adversity Adaptation and Genetic Improvement in Cold and Arid Regions of Inner Mongolia, Inner Mongolia Agricultural University, Hohhot 010018, China

**Keywords:** WRKY, *Caragana korshinskii*, genome-wide, evolutionary analysis, transcriptome

## Abstract

The WRKY transcription factor family plays a vital role in plant development and environmental response. However, the information of *WRKY* genes at the genome-wide level is rarely reported in *Caragana korshinskii*. In this study, we identified and renamed 86 *CkWRKY* genes, which were further classified into three groups through phylogenetic analysis. Most of these *WRKY* genes were clustered and distributed on eight chromosomes. Multiple sequence alignment revealed that the conserved domain (WRKYGQK) of the CkWRKYs was basically consistent, but there were also six variation types (WRKYGKK, GRKYGQK, WRMYGQK, WRKYGHK, WKKYEEK and RRKYGQK) that appeared. The motif composition of the CkWRKYs was quite conservative in each group. In general, the number of *WRKY* genes gradually increased from lower to higher plant species in the evolutionary analysis of 28 species, with some exceptions. Transcriptomics data and RT-qPCR analysis showed that the *CkWRKYs* in different groups were involved in abiotic stresses and ABA response. Our results provided a basis for the functional characterization of the CkWRKYs involved in stress resistance in *C. korshinskii*.

## 1. Introduction

Due to their static nature, plants must adapt to complex and constantly changing environmental conditions such as drought, salinity, alkalinity and temperature extremes. These environmental factors have profound effects on plant growth and yield [1,2]. Hence, it is pivotal to investigate the molecular mechanisms through which plants respond to stress. Recent research indicates that transcription factors (TFs) play important regulatory functions in plant stress response, for example, genes encoding TFs such as bHLH, MYB, NAC, WRKY and ERF have been found to be up-regulated in wheat under stress conditions [3].

WRKY is one of the largest TF families in plants and plays an important role in the transcriptional regulation of plants in response to a variety of environmental stresses [4,5,6]. They are named as such because of the highly conserved WRKY domain, containing 60 amino acids and a zinc-finger motif (C-X4-7-C-X22-23-H-X-H), and are classified into three groups according to the number of WRKY domains at the amino terminus and the types of zinc-finger motif at the carboxyl terminus. Group I has two WRKY domains and a C2H2 zinc-finger motif, group II has one WRKY domain and a C2H2 zinc-finger motif and group III has one WRKY domain and a C2HC zinc-finger motif. The group II WRKYs can be further divided into five subgroups, from IIa to IIe, based on other conserved amino acid sequences other than the WRKY domain [7]. WRKYs respond to environmental stress by binding to promoters of plant genes through the cis-element, with DNA sequence C/TTGACC/T, termed as W-box [8].

A large number of previous reports have demonstrated that WRKY family members regulate plant responses to various stresses and phytohormones [4,9]. In 1994, the first *WRKY* gene, *SPF1*, was identified from sweet potato [10]. With the completion of plant genome sequencing, more and more *WRKY* genes have been reported at the whole genome level in different species, such as *Arabidopsis thaliana*, *Lilium longiflorum* [11], *Daucus carota* [12], *Gossypium raimondii* and *G. hirsutum* [13] and *Musa acuminata* [14]. It was confirmed that *BHWRKY1* enhanced dehydration tolerance in transgenic *Nicotiana tabacum* [15]. SbWRKY30 regulated the expression of *Sbrd19* through binding to the W-box of its promoter to affect the drought tolerance of sorghum [16]. *CmWRKY17* overexpression in *A. thaliana* decreased its tolerance to salt compared to wild type [17]. OsMADS57 and its interacting OSTB1 synergetically activated the transcriptional regulation of *OsWRKY94* at low temperatures [18] and WRKYs also helped plants cope with reactive oxygen species [19], UV [20] and so on. In terms of phytohormones, *OsWRKY45* was up-regulated by ABA [21]; AtWRKY12 and AtWRKY13 were involved in the GA signaling regulation of plant flowering time [22]; *PqWRKY1* was induced by MeJA [23]; AtWRKY46, AtWRKY54, as well as AtWRKY70, were involved in both BR-regulated plant growth and drought response [24].

*Caragana korshinskii*, a leguminous shrub that predominantly grows in arid and semi-arid areas of China, is very tolerant to a wide range of environmental factors, such as drought, salt and alkali. To adapt to these adverse environments, it has evolved some stress resistance mechanisms. However, little information is available on WRKY TFs at the genomic level in *C. korshinskii*. Based on the genome data sequenced by our group and collaborators, we conducted a distribution, re-nomenclature, classification, conserved domain and motifs analysis of all CkWRKYs for the first time and explored the *CkWRKY* genes evolution in different plant species. This study will shed light on understanding the stress resistance mechanism of *C. korshinskii* via WRKYs by combining the transcriptome data under different stresses.

## 2. Results

### 2.1. Identification and Chromsome Distribution of WRKY Genes from C. korshinskii

To identify the WRKY encoding genes in *C. korshinskii*, we searched the sequences of *WRKY* genes from the draft genome data of *C. korshinskii* and the transcriptome data of *C. intermedia* (a closely related species of *C. korshinskii*) from previous studies [25]. After removing the incomplete and redundant sequences, a total of 86 putative *WRKY* genes were identified from the draft genome of *C. korshinskii*, which were designated as *CkWRKY1* to *CkWRKY86* based on their chromosomal distribution (Figure 1, Table 1 and Appendix A). Among the 86 genes, 80 genes (from *CkWRKY1* to *CkWRKY80*) were almost evenly distributed on the seven chromosomes and had nothing to do with the length of the chromosomes. There was a cluster distribution of some genes on one chromosome, which was similar to other species [26]. In addition, group II *WRKY*s existed in each chromatin. Only one gene (*CkWRKY81*) was located on the eighth chromosome and five genes (*CkWRKY82*~*CkWRKY86*) could not be found to have an association with any chromosome due to genome assembly problems.

The physical and chemical properties showed that the number of amino acids of these CkWRKYs varied between 139 and 744 aa, the relative molecular weight (MW) was between 15.44 and 81.33 kDa and the isoelectric point (pI) spanned from 4.82 to 10.14, according to the ExPASy online software. All of these CkWRKYs were predicted to be located in the nucleus by subcellular localization online software (Plant-mPLoc).

### 2.2. Phylogenetic and Multiple Sequence Alignment of the CkWRKYs and AtWRKYs

To classify the 86 CkWRKY proteins, an unrooted phylogenetic tree using the WRKY domain of the 86 CkWRKYs and 72 AtWRKYs was constructed using the Neighbor Joining (NJ) method (Figure 2 and Appendix A). According to the classification and phylogenetic analysis of the WRKYs from *A. thaliana*, the 86 CkWRKYs were divided into three groups (Figure 2 and Table 1). Group I contained 16 members, and 15 of them contained two conserved WRKY domains, which were further designated as the N-terminal WRKY domain (IN) and the C-terminal WRKY domain (IC); however, CkWRKY73-IN and AtWRKY10-IC contained only one domain [27]. It had been reported that some group I WRKYs contained three or four WRKY domains [28,29], which indicated that the number of conserved WRKY domains varied in group I in different species. Fifty-six members were clustered in group II, which possessed only one WRKY domain, and could be further divided into IIa-IIe subgroups. In addition, 14 WRKYs with one WRKY domain were identified in group III. The group I and group II CkWRKYs harbored a C2H2 zinc finger motif, while group III had a C2HC zinc finger motif, which was consistent with WRKY family members in other plant species. 

Sequence alignment analysis of the 86 CkWRKYs showed that each conserved domain had about 60 aa residues, including the highly conserved WRKYGQK motif and the zinc-finger motif. Previous studies have shown that the mutation of this conserved WRKY domain and changes to the zinc-finger structure had an impact on their binding activity [30,31,32].

Among the amino acid sequences of the 86 CkWRKYs, the highly conservative WRKYGQK motif had been found in 76 sequences, while the remaining ten CkWRKYs had six types of mutations (i.e., WRKYGKK, GRKYGQK, WRMYGQK, WRKYGHK, WKKYEEK and RRKYGQK). Among the 16 WRKYs in group I, the WRKY domain of two members (CkWRKY73 and CkWRKY70-IN) were mutated into WKKYEK and RRKYGQK, respectively. In group II, six CkWRKYs had mutations in the WRKY domain; these were CkWRKY13 in subgroup IIa (with GRKYGQK motif) and four members, including CkWRKY20, CkWRKY30, CkWRKY31 and CkWRKY37, in subgroup Iic (with WRKYGKK motif). In addition, CkWRKY30 and CkWRKY3 also had deletion in their zinc-finger structure. Another member of the Iic subgroup, CkWRKY52-Iic, lost its zinc-finger completely. The WRKY domain of the CkWRKY33 in subgroup Iie was mutated into WRMYGQK. The WRKY domain of two members, CkWRKY63 and CkWRKY80, in group III had been mutated into WRKYGHK and WRKYGKK, respectively. In order to understand the variation of the conserved WRKY domain in the different groups of CkWRKYs, the number of variant proteins was compared with the number of total members in the same groups to obtain the variation rate. The variation rate was 12.5% in group I, 10.71% in group II and 14.29% in group III. In addition, half of the mutated conserved motif was WRKYGKK (Figure 3, Appendix A).

### 2.3. The Conserved Motif Analysis 

As the conservative motif was sometimes related to the function of WRKY proteins [33], we used MEME to predict the conserved motifs of the CkWRKYs to better characterize this gene family. In our case, overall, 15 motifs were defined from the 86 CkWRKY proteins (Figure 4). Among them, the distribution of motifs 1 (which had the conserved WRKY domain sequences) and 2 were highly conserved in 85 CkWRKYs, while CkWRKY73, which lacked the C-terminal WRKY domain, only had motif 1. Moreover, the conserved motifs of the same group of CkWRKY proteins were basically the same, but there were also slight differences.

Among the 16 group I CkWRKYs, nine motifs (1–6, 9, 10 and 15) were identified. CkWRKY73 only contained motifs 1 and 4 due to the lack of the C-terminal WRKY domain sequence. CkWRKY54 contained five motifs (1–4, 6). CkWRKY7 also contained five motifs (1–5). CkWRKY77 and CkWRKY85 contained six motifs (1–6). CkWRKY12, 17, 45, 47, 50 and 72 contained seven motifs (1–6 and 10). CkWRKY83 and CkWRKY69 also contained seven motifs (1–6 and 15), while CkWRKY16 and CkWRKY84 contained eight motifs (1–6, 10 and 15) and CkWRKY70 contained all nine motifs. The above results indicated that motifs 1 and 4 were indispensable for the group I CkWRKYs.

In the 56 group II CkWRKYs, subgroup IIa had seven CkWRKYs, with seven kinds of motifs (1, 2, 4, 5, 7, 8 and 9) appearing. With the exception of CkWRKY58, which contained all of the motifs, the remaining six CkWRKYs exhibited motif deficiency and motifs 1, 2 and 4 were present in all of the Iia sequences. Subgroup Iib had 14 CkWRKYs, with eight kinds of motifs (1, 2, 4, 5, 7, 8, 9 and 13) appearing. With the exception of CkWRKY9, which lacked motif 8, CkWRKY46, which lacked motif 5, and CkWRKY40, which lacked motifs 5 and 13, the remaining 11 CkWRKYs all contained these eight kinds of motifs. There were 16 CkWRKYs in Iic, with a total of four motifs (1, 2, 4 and 5) found, nine CkWRKYs contained all four of these motifs, except that CkWRKY51 and CkWRKY52 lacked motifs 4 and 5 or 2, and CkWRKY2, 21, 30, 31 and 36 lacked motif 4. There were seven CkWRKYs in Iid, with four motifs (1, 2, 4 and 11) appearing; with the exception of CkWRKY 26, 32 and 53 lacking motif 4, the rest of the CkWRKYs contained all four motifs. There were 12 CkWRKYs in Iie, with four motifs (1, 2, 4 and 5) found; eight members, including CkWRKY4, 23, 48, 33, 55, 66, 79 and 82 lacked motif 5, and the remaining four members contained all four motifs. In summary, among the group II CkWRKYs, Iib had the most diverse motifs, and the motifs in the other subgroups (Iia, Iic and Iie) could be found in Iib. While Iid had the special motif 11, the motif composition of the Iid and Iie subgroups were basically similar, although the arrangement was slightly different. For example, the seven WRKYs of Iid subgroup all had motifs 1, 2 and 11, and their arrangements were completely consistent. In addition, we found that, with the exception of the absence of motif 4 in Iic and Iid, all of the other subgroups contained motifs 1, 2 and 4, indicating that motifs 1 and 2 were conserved in group II, while motif 4 may have emerged late and may play a role in the functional changes of different subgroups. Taken together, the above results indicated that there were more conserved motif changes in the group II WRKYs, suggesting that there might also be more changes in their function.

In the 14 group III CkWRKYs, seven motifs (1, 2, 4, 5, 9, 12 and 14) appeared. These CkWRKYs all contained motifs 1 and 2, while CkWRKY75 also contained motif 13, CkWRKY49 also contained motif 4, CkWRKY80 also contained 5, CkWRKY3, 18, 34, 61, 62 and 63 all contained motif 12, CkWRKY1 and 19 contained motifs 5 and 12 and CkWRKY42, 43 and 74 also contained motifs 5, 12 and 14. 

In summary, the results of the motif analysis for the different groups of CkWRKYs showed that motifs 1 and 2 existed simultaneously and were closely adjacent in all CkWRKYs, indicating that these two motifs might be the most primitive and played an indispensable role in the basic function of CkWRKYs. In addition, a pair of closely adjacent motifs, 3 and 6, were found, which only existed simultaneously in all members of the group I genes. According to the currently existing hypothesis, the group I WRKY proteins emerged first, followed by group II and III proteins [34]; it could be inferred that motifs 3 and 6 disappeared during evolution.

### 2.4. Evolution Analysis of WRKYs among Different Species

To understand the evolution of CkWRKYs in plant lineage, 27 more representative plant species were selected according to the complexity of the genome in the database (PlantTFDB). In this study, 1896 complete sequences of WRKY conserved domains were obtained from 28 species (including *C. korshinskii*) of plants spanning from algae, mosses, ferns, gymnosperms to angiosperms) (Figure 5).

Among all of the selected algae, the number of WRKYs was the smallest, while the number of WRKYs in angiosperm was relatively large. However, there were exceptions; for example, the angiosperm *Amborella trichopoda* had 29 WRKYs, which was less than the gymnosperm *Pseudotsuga menziesii* with 70 WRKYs and also less than *Sphagnum palustre* with 37 WRKYs. The 64 WRKYs of angiosperm *Epipactis helleborine* were also less than those of *Pseudotsuga menziesii*. In addition, the 15 WRKYs of Fern *Selaginella moellendorffii* were also less than that of *Sphagnum palustre*.

In general, the number of WRKYs varied greatly from lower to higher plants, ranging between 1 and 175. In algae, *Chlamydomonas reinhardtii* and *Dunaliella parva* contained one and five WRKYs, respectively. Among the mosses, 14 and 37 WRKYs were found in *Marchantia polymorpha* and *Sphagnum palustre,* respectively. In fern, *Selaginella moellendorffii* contained 15 WRKYs. Gymnosperms contained 9–70 WRKYs, whereas angiosperms contained 29–175 WRKYs.

As *C. korshinskii* belongs to leguminous plants, nine leguminous species were selected for comparison with *C. korshinskii* here (Figure 5). The results showed that the number of WRKYs in leguminous plants had little change, except that *Glycine max* had 175 WRKYs due to a WGD event [35]. The number of WRKYs in other legumes was within the range of 77–100 and *C. korshinskii* contained 86 WRKYs, which was relatively in the middle.

In order to explore the evolutionary relationship of WRKYs in plants, the WRKYs from 28 plant species were divided into seven categories (I, IIa, IIb, IIc, IId, IIe and III) based on their homology with *A. thaliana* (Figure 5 and Figure 6a). Based on the previous WRKYs origin hypothesis, group I appeared first, and then evolved into groups II and III [34]. We found that the aquatic algae only contained group I WRKYs, but group II and III began to appear in terrestrial plant mosses. The result was basically consistent with the above hypothesis. Although *Picea glauca* had no group I WRKYs, when we checked the initial download sequence, we did find that it contained one sequence with an incomplete WRKY domain. In addition, we also noticed that subgroup IIa did not begin to appear in the angiosperm *Epipactis helleborine*, while subgroups IIb, IIe and III had already appeared in bryophytes. There were zero in the gymnosperms *Pinus taeda* and *Picea sitchensis*, which might indicate that group II appeared earlier than III. Group II genes appeared in terrestrial plants and had occupied a dominant position in terms of quantity, among which the group II genes of *Oryza sativa* accounted for at least 48.45% and *Picea glauca* for at most 92.31% in all of the *WRKY* genes. Subgroups IIb and IIc again accounted for the majority of group II WRKYs, in which IIb accounted for a relatively high proportion, varying between 26% and 39% at the beginning, and then significantly decreasing to 8% in *Picea glauca*; meanwhile, the proportion of Iic was relatively stable, with at least 22% in *Marchantia polymorpha* and at most 58% in *Triticum urartu* (Figure 5 and Figure 6b).

Additionally, in the process of the sequence analysis, we found that the WRKYs sequence contained not only the WRKY conservative domain, but also some other domains (Figure 6c,d). In addition to algae and *Selaginella moellendorffii*, other terrestrial plants contained the Plant_zn_cluster domain [29,36]. Further analysis found that the Plant_zn_cluster domain was accompanied by the IId subgroup. In addition, we also found some other types of domains in the WRKYs sequence, such as the bZIP, WD40 and NAM domains. Since the other domains appeared in the WRKYs sequence, it was speculated that these minority domains might play a regulatory role in the coordination with the conserved WRKY domains. Among the 28 species, at least one kind of other domain (such as Plant_zn_clust) appeared, with the exception of *Chlamydomonas reinhardtii*, *Dunaliella parva* and *Selaginella moellendorffii*. In addition, *Phyllostachys edulis*, *Populus euphratica* and *Trifolium pratense*, with the largest number of other domains (as many as seven kinds), were all angiosperms. Furthermore, it was observed that *Populus euphratica* had the largest number of other domains, while the number of WRKYs in *Populus euphratica* was not the largest, indicating that the number of other domains was not positively correlated with the number of genes. Moreover, because *Populus euphratica* is a typical drought-resistant species in the northwest region, it was speculated that these extra domains might be related to its drought resistance.

### 2.5. Expression Profiles of CkWRKYs under Drought, Salt, Alkali and ABA Treatments

To understand the roles of the *CkWRKY* gene family under drought, salt, alkali and ABA treatment, we analyzed their expression profiles using the transcriptome data (Log2FC ≥ 1/FC ≥ 2). As shown in Figure 7a,b, the expression of *CkWRKY72-I* was up-regulated under all four treatments; *CkWRKY17-I* and *CkWRKY50-I* were up-regulated under drought, high pH and ABA treatment; *CkWRKY35-IIe* and *CkWRKY69-I* were up-regulated under high pH and ABA treatment; *CkWRKY10-Iic* and *CkWRKY59-Iib* were up-regulated under drought and salt treatment; *CkWRKY47-I* was up-regulated under salt and down-regulated under high pH treatment. The expression of the remaining 15 genes—*CkWRKY64-Iib*, *CkWRKY15-Iib*, *CkWRKY39-Iib*, *CkWRKY29-Iia*, *CkWRKY52-Iic*, *CkWRKY28-Iia*, *CkWRKY19-III*, *CkWRKY30-Iic*, *CkWRKY38-Iic*, *CkWRKY24-Iic*, *CkWRKY34-III*, *CkWRKY76-Iic*, *CkWRKY67-Iib*, *CkWRKY41-Iib* and *CkWRKY77-I—*changed only under drought stress. From the diagram (Figure 7b), it was found that some genes responded to multiple treatments. Due to the function of ABA in the stress pathway, the genes that responded to both stress and ABA were counted. For example, five *WRKY* genes (*CkWRKY17*, *35*, *50*, *69*, *72*) under drought, NaCl and high pH treatment overlapped with the genes under ABA treatment, indicating that these genes might exercise a stress-related function through ABA signal pathway.

In order to understand the response of *CkWRKY* genes to different stresses, these differentially expressed genes (DEGs) were classified into various groups (Figure 7c); we found that the expression level of the group I genes changed under all four treatments and the expression level of the group II genes changed under two treatments, while the group III genes changed only under one treatment, indicating that the group I genes played a core role in the plant stress response, followed by the group II genes and finally by the group III genes. In addition, the evolutionary analysis (Figure 5a) showed that the group I genes were the most primitive ones of the *WRKY* gene family and group II and group III had originated from group I, which might show that the more primitive the genes are, the more critical their function.

To analyze the DEGs in different groups under different treatments, the *CkWRKYs* that changed after four treatments were analyzed. As shown in Figure 7a–d, under drought treatment, a total of 20 DEGs were detected in the transcriptome, including four in group I, fourteen in group II and two in group III; under salt treatment, a total of four genes were changed, including two in group I and two in group II; under high pH treatment, seven genes were affected, including five in group I and two in group II; under ABA treatment, a total of five genes were changed, including four in group I and one in group II. DEGs were mainly clustered around the same branch. For example, the group I differential genes in the transcriptome were clustered together, such as *CkWRKY77* and *CkWRKY17*, as shown in Figure 7a, indicating that similar genes might have similar functions. Among the 16 group I genes in the genome of *C. korshinskii*, six played a role under four stresses, accounting for 37.5%. For the 56 group II genes, 16 played a role in these stresses, accounting for 28.57%. Among the 14 group III genes, two were responsive to these stresses, accounting for 14.29%. In addition, the Iib genes (42.86%) in group II were the most prevalent in all groups.

The above results show that group I played an Important role In different stresses. The group I genes were clustered together in the phylogenetic analysis, indicating that these genes might possess basic functions because they were the earliest genes of origin. In addition, although the conserved motifs of the group I members were basically the same, their functions were divergent, indicating that other sequences of these group I members might also play an important role.

In order to verify the reliability of the transcriptome results, RT-qPCR experiments were carried out (Figure 8). The results showed that these 24 DEGs had different degrees of expression variation compared with the control under the corresponding treatment, indicating that the transcriptome results had certain reliability. However, the situation was not very consistent, especially under drought treatment.

To investigate whether these 24 DEGs possessed interactions, homologous modeling with *A. thaliana* was used. The interaction network diagram of the DEGs under stress and ABA treatment is shown in Figure 9. Among the 24 genes, 17 were connected, indicating that these genes might interact. In addition, seven genes might not participate in the interaction because they existed alone in the network. In addition, it was predicted that the maximum number of lines of the interactive network connections was eight with CkWRKY17 and CkWRKY50. In addition, through the statistics of the number of lines for different groups of genes, it was found that there were no obvious rules. Taking six group I genes as an example, three genes (CkWRKY17, 50 and 72) had six lines, two genes (CkWRKY47 and 69) had one line and one gene (CkWRKY77) had zero lines. After observing the relationship between these six genes, it was found that only CkWRKY47 and CkWRKY69 had a predicted interaction. This result showed that the genes within the same group did not have similar and related interaction rules.

## 3. Discussion

Growing in the arid and semi-arid regions in Northwest China, *C. korshinskii* possesses drought resistance and adaptability to saline-alkali soil. Studying its adaption mechanism to these environmental factors will lay the foundation for molecular genetic breeding in the future. Previous studies have shown that WRKYs play an important role in stress response in plants, for example, AtWRKY1 regulates stomatal movement in *A. thaliana* under drought stress [37]; inactivated OsWRKY5 transcription factor enhances the drought tolerance of its mutants through the ABA signaling pathway [38]; GhWRKY1-like mediates drought tolerance in *A. thaliana* by regulating ABA biosynthesis [39]; the overexpression of *CmWRKY17* increases the salt sensitivity of transgenic plants in both in *Chrysanthemum morifolium* and *A. thaliana* [17]; the overexpression of *PtWRKY39* enhances the salt and alkali tolerance of transgenic tobacco [40]. In our research, through the analysis of the genome and transcriptome results, we evaluated the evolutionary status and structural characteristics of CkWRKYs in different groups among 28 species and found that multiple *CkWRKYs* genes were responsive to different environmental factors. Finally, through the analysis of two omics data, we found that the group I *CkWRKY*s possessed the most stress-related genes, suggesting that this group is closely related to stress response, which laid a molecular foundation for the functional research of this group. In brief, our research might provide essential information for understanding the stress tolerance of *C. korshinskii* at the genetic level.

WRKYs play an important regulatory role in various stresses [4,5], as well as in development [41,42]. With the development of genome sequencing technology, more and more WRKYs in legume species have been found, such as in *G. max* [43,44], *Medicago sativa* and *M. truncatula* [45,46]. In this study, the *WRKY* genes of *C. korshinskii* were investigated at the genome level for the first time. Eighty-one genes out of the full-length *CkWRKYs* were identified and located in eight chromatins; the other five might not be defined in chromatin due to assembly problems, which has been reported in other studies, such as in *Andrographis paniculata* [27]. In addition, it was observed that most of the *CkWRKY* genes showed a cluster distribution in chromatins, which had also been reported in other legume studies [26]. In the previous research on *CiWRKYs* based on the transcriptome, only 28 full-length *CiWRKY* genes were defined [25], indicating that the genomic research was conducive to the definition and discovery of *CkWRKY* genes and laid a foundation for the later research of *CkWRKY* genes. 

As the nomenclature of CkWRKYs was mainly based on homology with their orthologs in *A. thaliana* and other legume species or the order of discovery in this plant, it is quite confusing that one *CkWRKY* gene might have different names. With the aid of the genome data, we renamed the 86 *CkWRKYs* according to their distribution on the chromosomes and avoided the dilemma of several *CkWRKYs* having one *AtWRKY* ortholog (Table 1). The published *CkWRKYs/CiWRKYs* and their corresponding new names are summarized in Appendix A.

Phylogenetic analysis showed that the WRKY TFs were divided into three groups (group I, II and III), and group II was further divided into subgroups IIa-IIe, of which IIc had the largest number of WRKYs. These results were consistent with those obtained by other researchers [26,46,47,48,49,50]. In previous studies on CkWRKYs, the WRKYGQK conserved motif of one protein sequence was replaced by WKKYEEK, and the WRKYGQK of three protein sequences was replaced by WRKYGKK [25]. In our study, the WRKYGQK conserved domain had six types of mutations (WRKYGKK, GRKYGQK, WRMYGQK, WRKYGHK, WKKYEEK and RRKYGQK) in ten protein sequences, of which WRKYGKK accounted for half of all mutations, which showed that the number and types of conserved domains increased with the identification of more genes in the genome. The greatest conserved domain variation was found in Iic, and these results have also been reported in other species [26], suggesting that the WRKY proteins of IIc might have a variety of biological functions through the conserved domain variation. Motif analysis showed that some motifs were unique to groups I and III, which might lead to the specificity of the gene function of the two groups, and compared with the other groups, IIc only contained up to four conserved motifs. In addition, nine of the sixteen sequences contained these four motifs, which indicated that the motif of the same subgroup was highly conserved; similar conclusions were drawn in the study of *Cajanus cajan* [51]. 

Previous studies have shown that WRKY is one of the largest transcription factor families in plants, and the main research focused on higher plant WRKY proteins that have been detected in various organisms, such as spike mosses, single-celled green algae, slime molds and protozoa [7]. In monocots and dicots, such as rice, soybean, wheat and cotton, an especially large number of WRKY proteins with different functions have been confirmed in recent years [21,49,52,53,54]. However, our study was the first to identify and characterize WRKY proteins from the whole genome data of *C. korshinskii* and to carry out an evolution analysis from lower algae to higher monocotyledons, which had not previously been reported. Our study investigated WRKYs from the perspective of evolution, and the results showed that WRKY genes have been presented since green plants, and gene classification showed that group I genes were first presented in lower algae, and then group II and group III genes were gradually presented, indicating that the other subgroup genes might originate from group I. Some parts of these results were also reported in other studies [34,55,56,57], and our study enriched and validated the previous results and hypotheses. 

The present study found that WRKY genes have complex regulatory networks in abiotic stresses and phytohormone responses [4,5,58]. ABA is a phytohormone and plays essential roles in plant responses to abiotic stress [4]. In our study, drought, salt, high pH and ABA were selected to treat *C. korshinskii* seedlings, and samples were taken for the transcriptome sequencing. Subsequently, the differential *WRKYs* genes in the transcriptome data before and after treatment were analyzed and summarized and RT-qPCR verification was performed. A total of 24 DEGs were selected, and the expression analysis of these genes revealed different expression patterns for each *CkWRKY* under different treatments, providing a valuable resource for gene function research. Stress-related *WRKYs* were also found in the transcriptomes of other species, for example, *GaWRKYs* were salt-responsive [59], *CtWRKYs* responded to salt and cold stress [60] and *TaWRKYs* were responsive to drought [61]. The above results indicated that transcriptome sequencing played an essential role in the gene function research. In addition, the combination of transcriptome and genome analysis was helpful to understand the function of the genes in different subgroups in a more predictable way. In our study, among the DEGs under different treatments, the group I genes responded to a variety of stresses. In addition, the group II genes had the largest number, and subgroups IIb and IIc had the largest number in group II. Our study showed that the DEGs ratio was 16/86 in the group I *CkWRKYs*, which originated the earliest and responded to a variety of stress signals. The above results showed that although the number of group I genes was small, they played a key role in *C. korshinskii*, most likely because it appeared first. For example, CkWRKY50-I (formerly CkWRKY33) enhanced the drought tolerance in transgenic *A. thaliana* [62] and CiWRKY26, a homologous gene of CkWRKY72-I, responded to multiple stresses through expression pattern analysis and was located in the nucleus [25]. We also found that CkWRKY50-I showed a response to drought, and CkWRKY72-I was responsive to various stresses. In other species, studies have also shown that the *WRKY* genes in group I have multiple functions, for example, *GhWRKY3*-I could be induced by phytohormones and pathogens [63], NbWRKYs-I [64] were related to defense response and AtWRKY1-I played an important regulatory role in the drought, salt, light and nitrogen signaling pathways [37,65,66].

In conclusion, the definition of WRKY genes based on their genome, the study of gene phylogeny and classification analysis were conducive to the finding of more genes and the classifying of different types of genes. In addition, the evolutionary analysis of 28 different species was helpful to understand the origin relation of the WRKYs. Further, the application of the transcriptome was very helpful for gene function prediction, combined with the gene classification and evolutionary relationship.

## 4. Materials and Methods

### 4.1. Genome-Wide Identification and Distribution of CkWRKYs

To comprehensively identify the *CkWRKY* genes, the genome sequences were obtained from the *C. korshinskii* genome V1.0, which was completed by our team and Lanzhou University, and the transcriptome data from previous studies [25]. The NCBI Conserved Domains Database (CDD) was used to identify the WRKY domain, then the incomplete WRKY domain and the repetitive sequences were manually removed; finally, the *CkWRKYs* were identified. After this, ExPASy (http://web.expasy.org/compute_pi accessed on 1 March 2022) and Plant-mPLoc (http://www.csbio.sjtu.edu.cn/bioinf/plant-multi/ accessed on 1 March 2022) were used to predict the molecular weight (MW), isoelectric point (pI) and protein subcellular localization of the defined CkWRKY proteins. 

To locate the positions of *CkWRKY* genes on the chromosomes of *C. korshinskii*, the MapInspect Software (http://www.softsea.com/download/MapInspect.html accessed on 1 March 2022) was used to align the distribution of the putative *CkWRKYs* based on the genome annotation. 

### 4.2. Phylogenetic, Conserved Domains and Motif Analysis

The phylogenetic tree of *C. korshinskii* and *A. thaliana* was constructed using the truncated amino acid sequences containing a WRKY conserved domain using MEGA7.0 with the Neighbor Joining (NJ) method and the bootstrap test was set for 1000 replications. Multiple sequence alignment files were also obtained using MEGA7.0 to analyze the WRKY conserved domains, and the logo of these sequences was constructed using the WebLogo server (http://weblogo.berkeley.edu/logo.cgi accessed on 20 March 2022). MEME (http://meme.nbcr.net/meme/ accessed on 20 March 2022) analysis was performed to discover the conserved motifs of the full-length protein, with the number of displayed motifs set to 15 and the rest using default settings. Visualization and beautification of the phylogenetic tree, multiple sequence alignment and conserved motif diagram were performed using iTOL (https://itol.embl.de/ accessed on 25 March 2022).

### 4.3. Evolution Analysis

In order to understand the evolution of CkWRKYs in different species, WRKY family members from 27 other representative species from lower to higher were selected to compare with *C. korshinskii* for evolutionary analysis. These species were: *Chlamydomonas reinhardtii*, *Dunaliella parva*, *Marchantia polymorpha*, *Sphagnum palustre*, *Selaginella moellendorffii*, *Pseudotsuga menziesii*, *Pinus taeda*, *Picea glauca*, *Picea sitchensis*, *Amborella trichopoda*, *Epipactis helleborine*, *Zea mays*, *Oryza sativa*, *Phyllostachys edulis*, *Triticum urartu*, *Aquilegia coerulea*, *Arabidopsis thaliana*, *Populus euphratica*, *Arachis ipaensis*, *Cajanus cajan*, *Glycine max*, *Phaseolus vulgaris*, *Vigna angularis*, *Lotus japonicus*, *Cicer arietinum*, *Medicago truncatula* and *Trifolium pratense,* respectively. The WRKY sequences of these species were downloaded from PlantTFDB (http://planttfdb.gao-lab.org/family accessed on 1 April 2022), the repetitive sequences were manually removed, then the sequences without conservative WRKY domains were deleted through CDD analysis; the remaining sequences were used for comparative analysis. The evolution of 28 species (covering algae, mosses, ferns, gymnosperms and angiosperms) was analyzed using the TimeTree database (http://timetree.org/ accessed on 1 May 2022).

### 4.4. Abiotic Stress and ABA Treatments 

The seedlings of *C. korshinskii* were cultured in the growth chamber. The culture conditions were as follows: vegetative soil and vermiculite (V/V = 1:3), photorhythm 16/8 h (light/dark) and culture temperature of about 22 °C. When the seedlings had grown for 3 weeks, they was treated with drought (withholding water), salt (watered with 200 mM NaCl solution), high pH (watered with 200 mM NaHCO3 solution using NaOH to adjust the pH to 10) and ABA (Sprayed with 100 μM ABA solution), then the whole plant of each seedling was sampled at 0, 6 and 12 days, respectively, after drought, salt and ABA treatments, while samples under high pH treatment were collected in at 0, 3 and 6 days due to the premature yellowish color of the seedlings after treatment. The mixture of three seedlings was taken as one sample, and the experiments was repeated for three times. All samples were snap frozen with liquid nitrogen and stored at −80 °C for transcriptome sequencing and real-time quantitative PCR (RT-qPCR). 

### 4.5. Heatmap Analysis and RT-qPCR Verification of the Transcriptome Data

To check the response of *CkWRKYs* to abiotic stress and ABA, differentially expressed *CkWRKYs* with over 2-fold changes in the transcriptome data were screened, Heatmaps of these *CkWRKYs* were constructed using log_2_-based FPKM values and the results were visualized using TBtools. All of the selected genes were verified through RT-qPCR (using TB Green qPCR Master Mix (TaKaRa)) with a Roche LightCycler 480 system and the primer sequences were listed in Appendix A. The results of the RT-qPCR were calculated according to the 2^−ΔΔCT^ method [67] and were normalized using *CiEF1α* reference gene [GenBank No.: KC679842] [68]. All RT-qPCR assays were carried out with three technical replicates. 

## 5. Conclusions

In this study, we performed a genome-wide identification of *WRKY* genes in *C. korshinskii* and identified a total of 86 *CkWRKY* genes. These genes were classified into three groups—I, II and III—and we further classified the members in group II into five subgroups based on their phylogenetic relationships. The basic features, phylogenetic, conserved domain, motif and species evolution of these genes were analyzed, providing a foundational understanding of the evolutionary relationships within the *CkWRKY* gene family. The expression profile of the *CkWRKY*s was studied using the transcriptomic data, and the results revealed that 24 *CkWRKY*s were influenced by abiotic stresses and ABA. Our results provided a basis for the functional characterization of the *CkWRKY*s involved in stress resistance in *C. korshinskii*.

## Figures and Tables

**Figure 1 ijms-24-09519-f001:**
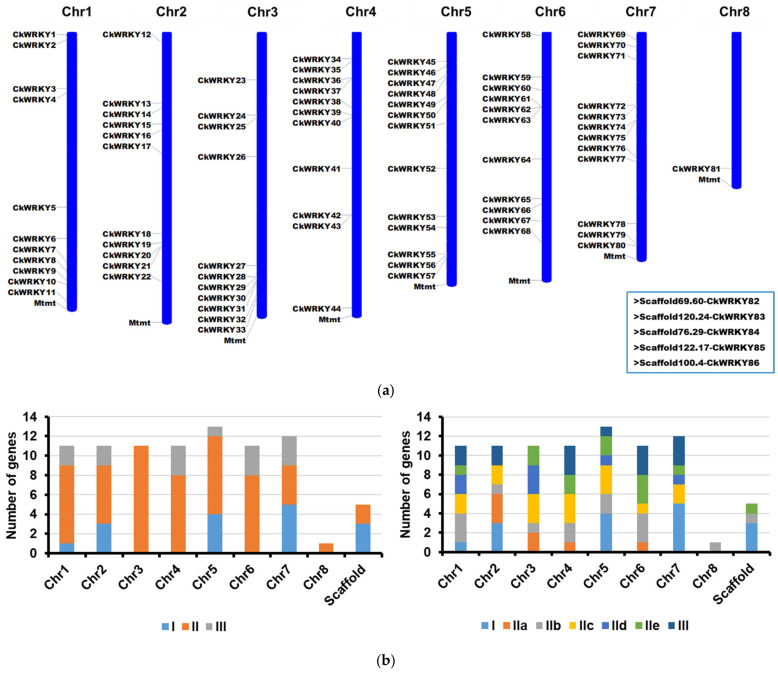
Distribution of *CiWRKY* genes on eight chromosomes of *C. korshinskii*. (**a**) Distribution of 81 *CkWRKY* genes on each chromosome and the five genes marked by (>) in the lower right corner indicated that they had no chromatin to be located. (**b**) The distribution and quantity statistics of different groups and subgroups of *CkWRKYs* on each chromosome.

**Figure 2 ijms-24-09519-f002:**
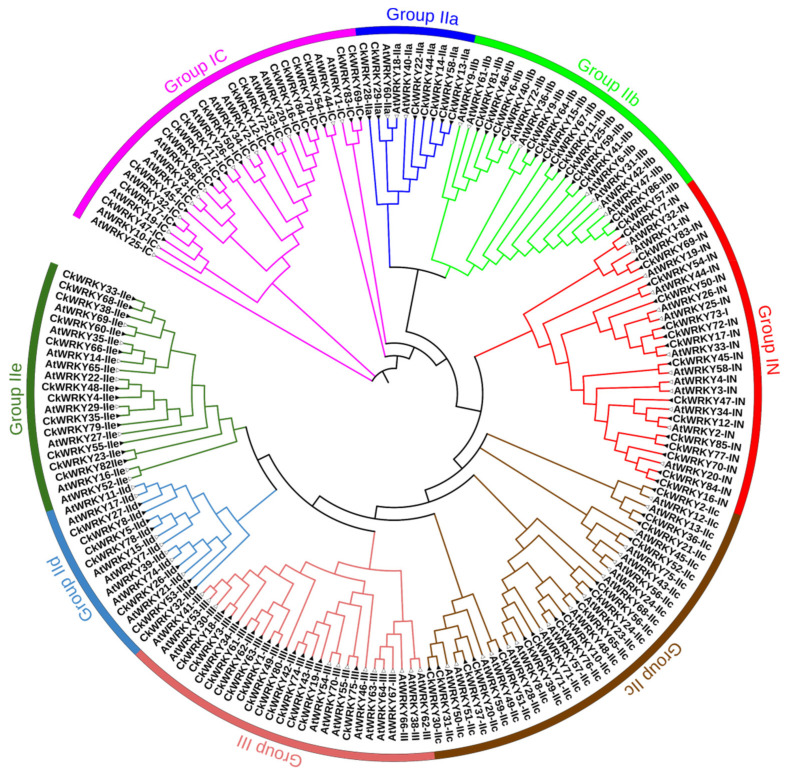
Phylogenetic analysis of 158 WRKYs containing the conserved WRKY domains from both *C. korshinskii* and *A. thaliana* using the NJ method. These WRKYs were clustered into eight groups or subgroups: IN, IC, IIa, IIb, IIc, IId, IIe and III. CkWRKYs were marked by the solid triangle and AtWRKYs were labeled with the hollow triangle.

**Figure 3 ijms-24-09519-f003:**
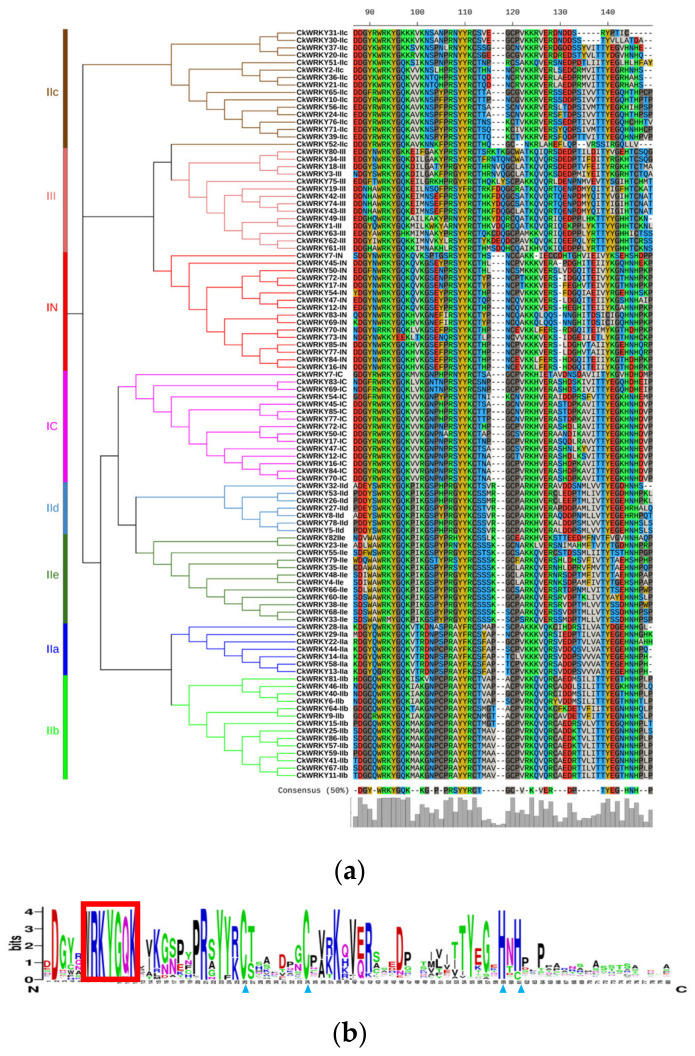
Amino acid sequence alignment and Logo diagram of the typical WRKY domains of *C. korshinskii*. (**a**) Alignment of the amino acid sequence. (**b**) Logo diagram of the conserved WRKY domain; the red box indicated WRKY domain and the blue triangle indicated zinc-finger structure.

**Figure 4 ijms-24-09519-f004:**
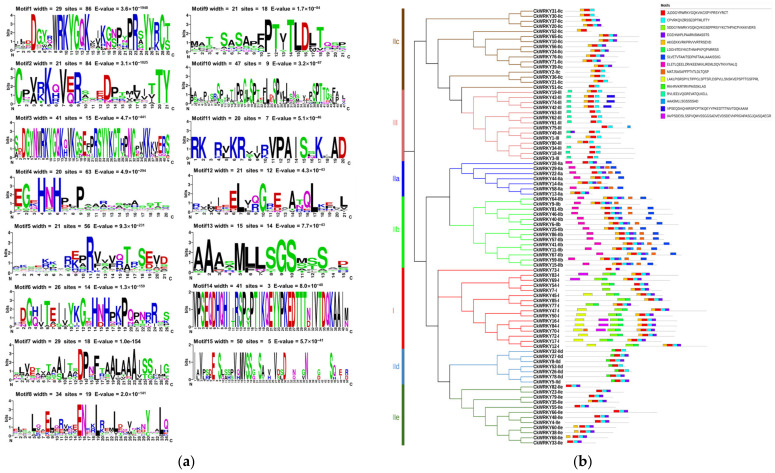
Conserved motifs in CkWRKY proteins. (**a**) Logo diagram of the conserved WRKY motif. (**b**) Conservative motif distribution in CkWRKYs. (**a**,**b**) Fifteen motifs were identified using MEME, as indicated by colored rectangles. The height of the rectangles was proportional to the-log (*p*-value), truncated at the height of a motif with a *p*-value of 1 × 10^−10^.

**Figure 5 ijms-24-09519-f005:**
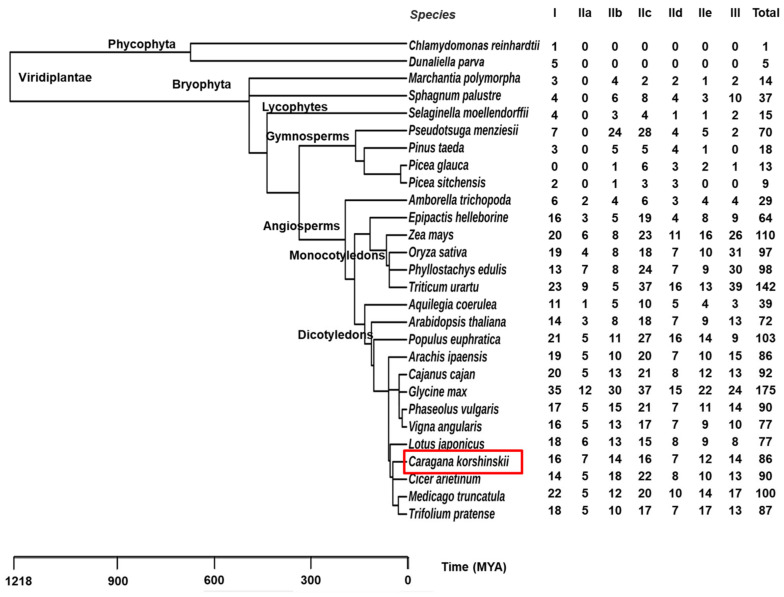
Evolutionary analysis WRKY proteins of 28 plant species by TimeTree database.

**Figure 6 ijms-24-09519-f006:**
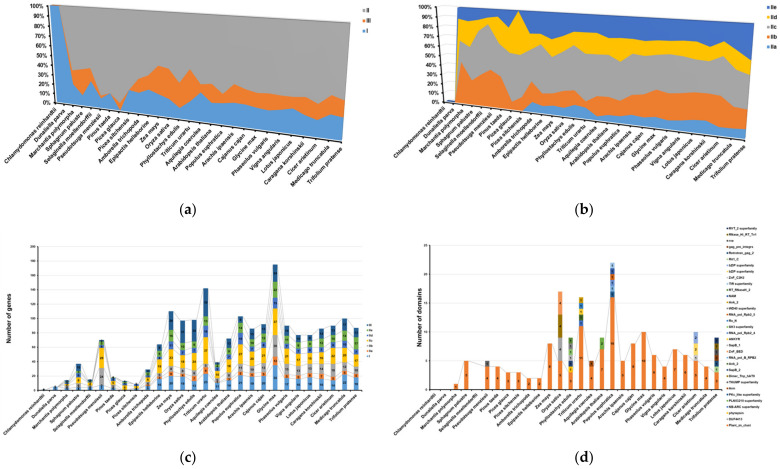
Statistics of the number and other domains of WRKY proteins in 28 species. (**a**) Statistics of the proportion and number of different groups and subgroups of WRKYs. (**b**) Statistics of the proportion and number of different subgroups of group II WRKYs. (**c**) Statistics of protein number in different groups and from different species. (**d**) Statistics of other domains of WRKY proteins in different groups and from different species.

**Figure 7 ijms-24-09519-f007:**
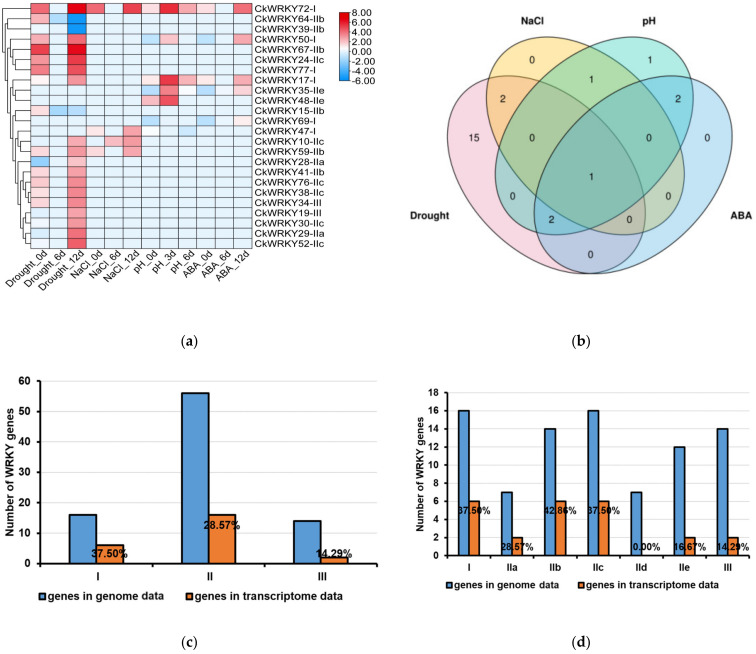
Expression profiles of *CkWRKY* genes under the treatment of drought, salt, pH and ABA. (**a**) Heatmaps showing the clustering of these genes were created using log_2_-based FPKM values, and the scale represented the signal intensities of the FPKM values. (**b**) The Venn diagram showed the number of overlapped *CkWRKY* genes under different treatments. (**c**–**e**) Proportion of differentially expressed *CkWRKYs* in various groups and subgroups.

**Figure 8 ijms-24-09519-f008:**
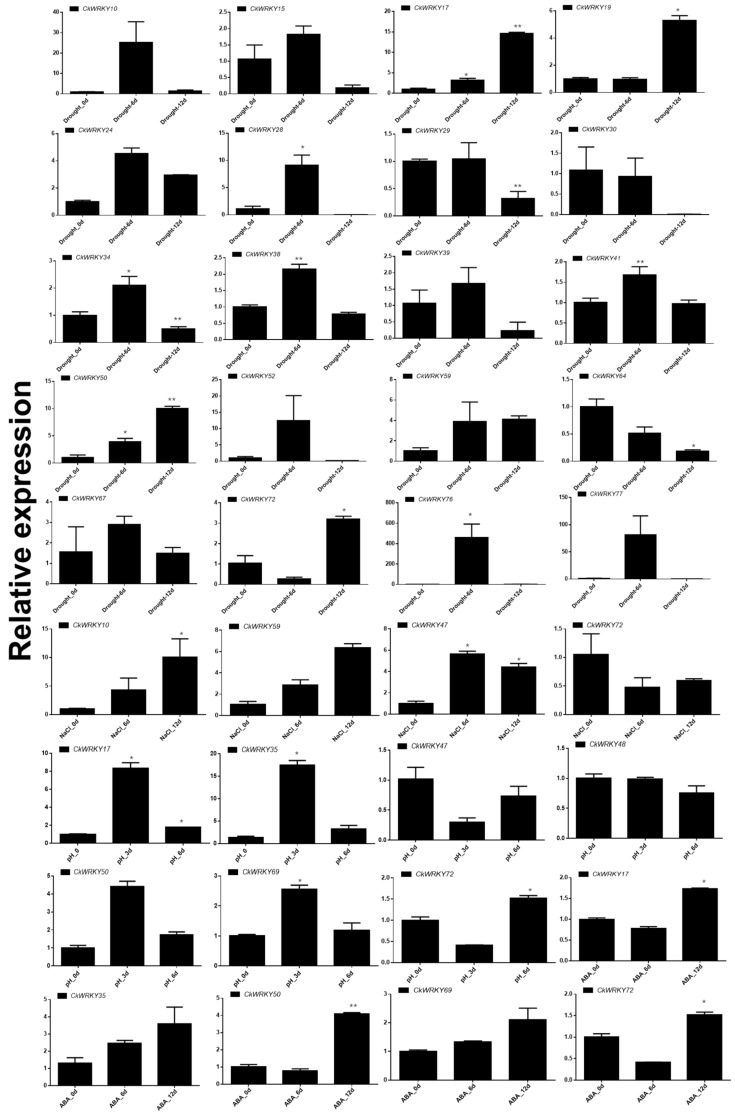
Expression profiles of selected *CkWRKYs* under treatment of drought, salt, high pH and ABA using RT-qPCR. Note: The results of the RT-qPCR were calculated according to the 2^−ΔΔCT^ method, and were normalized using *CiEF1α* as the reference gene, * means *p* < 0.05 and ** means *p* < 0.01, using Student’s *t*-test.

**Figure 9 ijms-24-09519-f009:**
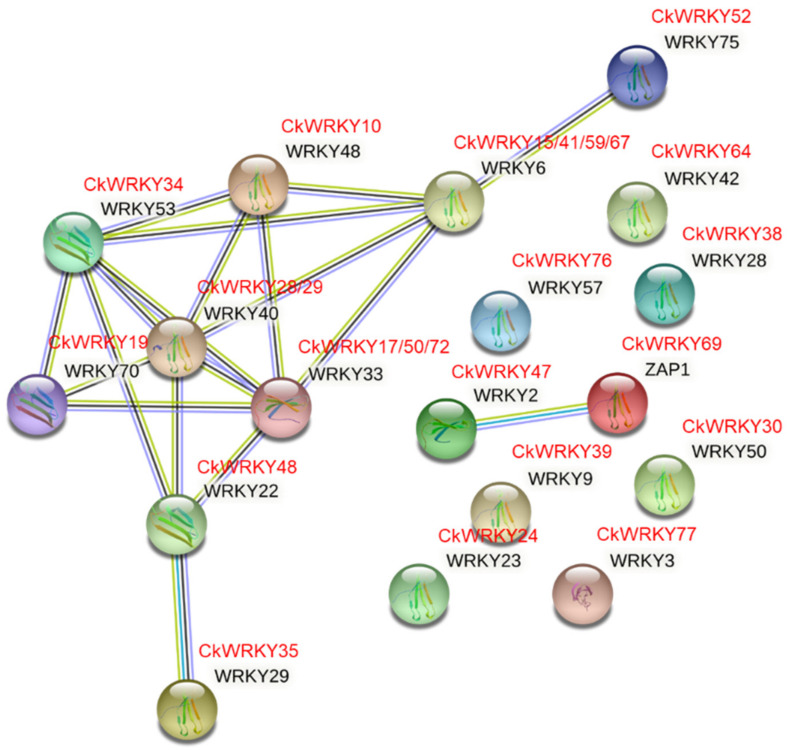
Interaction network pattern of 24 differential *CkWRKYs* under drought, salt, high pH and ABA treatments using STRING (https://cn.string-db.org/cgi/ accessed on 1 January 2020).

**Table 1 ijms-24-09519-t001:** The WRKYs identified from *C. korshinskii*.

Chromosome	Gene Name	ORF (aa)	pI	MW (kDa)	Conserved Heptapeptide	Zinc-Finger Type	Domain Number	Group/ Subgroup	Ortholog in *A. thaliana*
Chr1	*CkWRKY1*	292	5.94	33.77	WRKYGQK	C2HC	1	III	AtWRKY70
	*CkWRKY2*	232	7.11	26.32	WRKYGQK	C2H2	1	IIc	AtWRKY12
	*CkWRKY3*	361	5.37	40.77	WRKYGQK	C2HC	1	III	AtWRKY46
	*CkWRKY4*	333	5.72	36.37	WRKYGQK	C2H2	1	IIe	AtWRKY27
	*CkWRKY5*	336	9.67	36.59	WRKYGQK	C2H2	1	IId	AtWRKY15
	*CkWRKY6*	594	6.12	64.39	WRKYGQK	C2H2	1	IIb	AtWRKY72
	*CkWRKY7*	507	7.64	54.68	WRKYGQK/WRKYGQK	C2H2	2	I	AtWRKY32
	*CkWRKY8*	312	9.81	33.88	WRKYGQK	C2H2	1	IId	AtWRKY11
	*CkWRKY9*	391	8.98	42.96	WRKYGQK	C2H2	1	IIb	AtWRKY42
	*CkWRKY10*	383	6.15	41.86	WRKYGQK	C2H2	1	IIc	AtWRKY48
	*CkWRKY11*	504	5.54	54.71	WRKYGQK	C2H2	1	IIb	AtWRKY42
Chr2	*CkWRKY12*	223	8.16	24.85	WRKYGQK	C2H2	1	IIa	AtWRKY40
	*CkWRKY13*	144	5.45	16.16	GRKYGQK	C2H2	1	IIa	AtWRKY60
	*CkWRKY14*	744	5.77	80.62	WRKYGQK/WRKYGQK	C2H2	2	I	AtWRKY2
	*CkWRKY15*	482	6.08	52.90	WRKYGQK	C2H2	1	IIb	AtWRKY6
	*CkWRKY16*	587	6.48	63.72	WRKYGQK/WRKYGQK	C2H2	2	I	AtWRKY20
	*CkWRKY17*	506	7.69	56.17	WRKYGQK/WRKYGQK	C2H2	2	I	AtWRKY33
	*CkWRKY18*	368	5.35	41.30	WRKYGQK	C2HC	1	III	AtWRKY41
	*CkWRKY19*	313	4.88	35.23	WRKYGQK	C2HC	1	III	AtWRKY70
	*CkWRKY20*	205	7.72	23.28	WRKYGKK	C2H2	1	IIc	AtWRKY51
	*CkWRKY21*	250	8.84	27.87	WRKYGQK	C2H2	1	IIc	AtWRKY13
	*CkWRKY22*	299	7.11	33.66	WRKYGQK	C2H2	1	IIa	AtWRKY40
Chr3	*CkWRKY23*	307	6.05	33.80	WRKYGQK	C2H2	1	IIe	AtWRKY27
	*CkWRKY24*	314	7.17	35.06	WRKYGQK	C2H2	1	IIc	AtWRKY23
	*CkWRKY25*	429	8.9	47.18	WRKYGQK	C2H2	1	IIb	AtWRKY47
	*CkWRKY26*	350	9.69	39.23	WRKYGQK	C2H2	1	IId	AtWRKY21
	*CkWRKY27*	339	9.51	36.64	WRKYGQK	C2H2	1	IId	AtWRKY17
	*CkWRKY28*	260	8.19	29.15	WRKYGQK	C2H2	1	IIa	AtWRKY40
	*CkWRKY29*	275	7.03	30.97	WRKYGQK	C2H2	1	IIa	AtWRKY40
	*CkWRKY30*	175	5.49	20.40	WRKYGKK	-	1	IIc	AtWRKY50
	*CkWRKY31*	155	5.81	18.30	WRKYGKK	-	1	IIc	AtWRKY50
	*CkWRKY32*	336	9.57	37.09	WRKYGQK	C2H2	1	IId	AtWRKY7
	*CkWRKY33*	162	4.82	17.80	WRMYGQK	C2H2	1	IIe	AtWRKY65
Chr4	*CkWRKY34*	361	5.35	41.13	WRKYGQK	C2HC	1	III	AtWRKY41
	*CkWRKY35*	284	5.22	32.43	WRKYGQK	C2H2	1	IIe	AtWRKY27
	*CkWRKY36*	233	8.96	26.64	WRKYGQK	C2H2	1	IIc	AtWRKY13
	*CkWRKY37*	186	6.65	21.55	WRKYGKK	C2H2	1	IIc	AtWRKY51
	*CkWRKY38*	252	5.77	28.30	WRKYGQK	C2H2	1	IIe	AtWRKY65
	*CkWRKY39*	333	6.18	36.99	WRKYGQK	C2H2	1	IIc	AtWRKY71
	*CkWRKY40*	482	5.39	53.20	WRKYGQK	C2H2	1	IIb	AtWRKY9
	*CkWRKY41*	558	5.65	60.72	WRKYGQK	C2H2	1	IIb	AtWRKY6
	*CkWRKY42*	300	6.25	33.56	WRKYGQK	C2HC	1	III	AtWRKY70
	*CkWRKY43*	317	5.94	35.47	WRKYGQK	C2HC	1	III	AtWRKY70
	*CkWRKY44*	311	8.59	34.92	WRKYGQK	C2H2	1	IIa	AtWRKY40
Chr5	*CkWRKY45*	512	7.02	55.86	WRKYGQK/WRKYGQK	C2H2	2	I	AtWRKY4
	*CkWRKY46*	564	7.32	62.55	WRKYGQK	C2H2	1	IIb	AtWRKY72
	*CkWRKY47*	741	5.57	81.33	WRKYGQK/WRKYGQK	C2H2	2	I	AtWRKY20
	*CkWRKY48*	335	5.84	36.67	WRKYGQK	C2H2	1	IIe	AtWRKY27
	*CkWRKY49*	307	6.02	34.66	WRKYGQK	C2HC	1	III	AtWRKY70
	*CkWRKY50*	563	6.3	62.42	WRKYGQK/WRKYGQK	C2H2	2	I	AtWRKY33
	*CkWRKY51*	322	5.2	35.79	WRKYGQK	C2H2	1	IIc	AtWRKY49
	*CkWRKY52*	139	10.14	15.44	WRKYGQK	-	1	IIc	AtWRKY75
	*CkWRKY53*	318	9.62	35.76	WRKYGQK	C2H2	1	IId	AtWRKY21
	*CkWRKY54*	448	8.91	50.16	WRKYGQK/WRKYGQK	C2H2	2	I	AtWRKY44
	*CkWRKY55*	265	4.85	30.29	WRKYGQK	C2H2	1	IIe	AtWRKY35
	*CkWRKY56*	296	8.62	33.14	WRKYGQK	C2H2	1	IIc	AtWRKY23
	*CkWRKY57*	518	6.89	56.40	WRKYGQK	C2H2	1	IIb	AtWRKY42
Chr6	*CkWRKY58*	307	8.75	34.15	WRKYGQK	C2H2	1	IIa	AtWRKY40
	*CkWRKY59*	469	6.28	50.69	WRKYGQK	C2H2	1	IIb	AtWRKY6
	*CkWRKY60*	266	5.87	28.92	WRKYGQK	C2H2	1	IIe	AtWRKY69
	*CkWRKY61*	299	5.56	34.11	WRKYGQK	C2HC	1	III	AtWRKY70
	*CkWRKY62*	313	5.57	35.40	WRKYGQK	C2HC	1	III	AtWRKY70
	*CkWRKY63*	284	7.59	32.77	WRKYGHK	C2HC	1	III	AtWRKY70
	*CkWRKY64*	456	7.57	50.70	WRKYGQK	C2H2	1	IIb	AtWRKY42
	*CkWRKY65*	389	6.4	42.37	WRKYGQK	C2H2	1	IIc	AtWRKY48
	*CkWRKY66*	483	6.11	53.23	WRKYGQK	C2H2	1	IIe	AtWRKY35
	*CkWRKY67*	610	6.33	65.55	WRKYGQK	C2H2	1	IIb	AtWRKY6
	*CkWRKY68*	224	6.02	24.51	WRKYGQK	C2H2	1	IIe	AtWRKY65
Chr7	*CkWRKY69*	550	6.01	60.78	WRKYGQK/WRKYGQK	C2H2	2	I	AtWRKY1
	*CkWRKY70*	579	7.67	62.70	RRKYGQK/WRKYGQK	C2H2	2	I	AtWRKY20
	*CkWRKY71*	264	8.16	29.90	WRKYGQK	C2H2	1	IIc	AtWRKY71
	*CkWRKY72*	574	7.21	63.30	WRKYGQK/WRKYGQK	C2H2	2	I	AtWRKY33
	*CkWRKY73*	321	9.33	36.85	WKKYEEK	C2H2	1	I	AtWRKY33
	*CkWRKY74*	315	5.55	35.15	WRKYGQK	C2HC	1	III	AtWRKY70
	*CkWRKY75*	352	6.27	38.56	WRKYGQK	C2HC	1	III	AtWRKY55
	*CkWRKY76*	278	6.61	30.74	WRKYGQK	C2H2	1	IIc	AtWRKY57
	*CkWRKY77*	387	7.76	42.48	WRKYGQK/WRKYGQK	C2H2	2	I	AtWRKY3
	*CkWRKY78*	312	9.53	34.35	WRKYGQK	C2H2	1	IId	AtWRKY15
	*CkWRKY79*	252	8.67	28.53	WRKYGQK	C2H2	1	IIe	AtWRKY29
	*CkWRKY80*	292	6.71	33.07	WRKYGKK	C2HC	1	III	AtWRKY41
Chr8	*CkWRKY81*	554	6.83	60.54	WRKYGQK	C2H2	1	IIb	AtWRKY72
Scaffold	*CkWRKY82*	186	6.06	20.79	WRKYGQK	C2H2	1	IIe	AtWRKY27
	*CkWRKY83*	593	6.26	65.67	WRKYGQK/WRKYGQK	C2H2	2	I	AtWRKY1
	*CkWRKY84*	587	6.48	63.74	WRKYGQK/WRKYGQK	C2H2	2	I	AtWRKY20
	*CkWRKY85*	552	5.64	59.82	WRKYGQK/WRKYGQK	C2H2	2	I	AtWRKY3
	*CkWRKY86*	517	6.91	56.32	WRKYGQK	C2H2	1	IIb	AtWRKY47

## Data Availability

The data used to support the findings of this study are available from the corresponding author upon request.

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
