# Peer review of "Genome-Wide Analysis of WRKY Transcription Factors Involved in Abiotic Stress and ABA Response in Caragana korshinskii"

_ijms, 2023, doi:10.3390/ijms24119519_

Round 1
Reviewer 1 Report
In this manuscript, the authors perform bioinformatic analyses of the distribution and composition of WRKY transcription factor genes in Caragana korshinskii. The analyses seem relatively standard and the results presented appear to be supported by the data. However, the English language and presentation of the manuscript is very poor, making the presentation unclear. The manuscript requires extensive editing to correct language errors and to make the work presentable.
English language is poor and requires extensive editing.
Author Response
Dear editors,
Thank you and the reviewers very much for the comments on our manuscript entitled “Genome-Wide Analysis of WRKY Transcription Factors Involved in Abiotic Stress and ABA Response in Caragana korshinskii ”. All the suggestions are very valuable for improving our manuscript. We had modified it carefully according to the comments point-by-point (Please see the attachment). All changes were labeled in red color in the revised version. Hopefully now it is better for your consideration to publish in International Journal of Molecular Sciences.
Best regards,
Ruigang Wang

Reviewer 2 Report
Line 12: Please do not start the group word with an uppercase letter. It is not a proper noun. It had been used all uppercased throughout the manuscript. Please correct all.
Line 13: You already stated that you have carried out this study on C. korshinskii. Thus, the usage in line 13 is needless.
In lines 16-17: please coordinate the sentence in a fluent version such as: "In general, the WRKY genes gradually increased from low to high plant in the evolutionary analysis of 28 species."
Some minor fluency issues could be enhanced.
Author Response
Dear editors,
Thank you and the reviewers very much for the comments on our manuscript entitled “Genome-Wide Analysis of WRKY Transcription Factors Involved in Abiotic Stress and ABA Response in Caragana korshinskii ”. All the suggestions are very valuable for improving our manuscript. We had modified it carefully according to the comments point-by-point(Please see the attachment). All changes were labeled in red color in the revised version. Hopefully now it is better for your consideration to publish in International Journal of Molecular Sciences.
Best regards,
Ruigang Wang

Reviewer 3 Report
Caragana korshinskii is found naturally in sandy and desert areas of northern China and Mongolia, and its tolerance mechanisms to several stresses, including drought, are being investigated. Both physiological researches aimed at understanding the assessment and mechanisms of tolerance and molecular biology research aimed at identifying and analysing the function of genes involved in tolerance have been conducted and are expected to expand.
WRKY transcription factors are known to regulate various physiological phenomena in many plants. Still, due to many family genes, it is essential to organise information on many family genes to understand the whole picture.
This paper presents an informatics analysis of the WRKY genes in Caragana korshinskii, which has not been done before, to provide a complete picture. WRKY genes are essential transcription factors in various physiological processes, from plant development to stress tolerance. Furthermore, gene expression analyses were carried out during several stress exposures, and each WRKY gene was characterised by expression. The results of this paper may provide essential information for understanding stress tolerance in Caragana korshinskii at the genetic level.
The following are comments.
1) Subcellular localisation predictions in Table 1 are all Nucleus, and as there was no specific discussion in the text, this column is considered unnecessary.
2) Is it possible to use a notation in Figure 2 to easily distinguish between CkWRKY and AtWRKY? For example, change the colour of the letters.
3) The transcriptome analysis clearly shows the different uses of WRKY in Caragana korshinskii and is enjoyable to plant stress tolerance researchers. As this point is somewhat lacking in the discussion, a more specific description of the use of each WRKY gene and the characteristics of Caragana korshinskii would make the report more attractive to the reader.
Author Response

(The authors gave the same response as above.)

Round 2
Reviewer 1 Report
The authors have adequately addressed my concerns.